# Benefits of Two 24-Week Interactive Cognitive–Motor Programs on Body Composition, Lower-Body Strength, and Processing Speed in Community Dwellings at Risk of Falling: A Randomized Controlled Trial

**DOI:** 10.3390/ijerph19127117

**Published:** 2022-06-10

**Authors:** Hugo Rosado, Catarina Pereira, Jorge Bravo, Joana Carvalho, Armando Raimundo

**Affiliations:** 1Departamento de Desporto e Saúde, Escola de Saúde e Desenvolvimento Humano, Universidade de Évora, 7004-516 Évora, Portugal; clnp@uevora.pt (C.P.); jorgebravo@uevora.pt (J.B.); ammr@uevora.pt (A.R.); 2Comprehensive Health Research Centre (CHRC), Universidade de Évora, 7004-516 Évora, Portugal; 3Faculdade de Desporto, Universidade do Porto, 4099-002 Porto, Portugal; mjoanacarvalho@reit.up.pt; 4CIAFEL-Research Centre in Physical Activity, Health and Leisure, Universidade do Porto, 4200-450 Porto, Portugal

**Keywords:** aging, falls, psychomotor intervention, bone mineral density, cognitive function, muscle strength

## Abstract

This 24-week randomized controlled trial study evaluated the effects of two interactive cognitive–motor programs on body composition, lower-body strength, and processing speed in community dwellings at risk of falling. Forty-eight participants (75.0 ± 5.4 years) were allocated into EG1 (psychomotor intervention program), EG2 (combined program (psychomotor intervention + whole-body vibration)), and a control group. EG programs induced significant improvements in bone mass, lower-body strength, and processing speed (*p* < 0.05), with similar treatment effects on lower-body strength and processing speed and higher bone mineral content and density within EG2. The fall rate decreased in EG1 (44.2%) and EG2 (63%) (*p* < 0.05). After the 12-week no-intervention follow-up, improvements in lower-body strength were reversed in both EGs, but those in processing speed were maintained, mainly in EG2 (*p* < 0.05). In conclusion, both programs were accepted and well tolerated. The combined program led to additional benefits in bone mass. Both programs positively impacted physical and cognitive risk factors for falls and injuries. They induced similar improvements in lower-body strength and processing speed, decreasing the fall rate. These findings suggest that both programs are successful for fall and injury prevention in the studied population.

## 1. Introduction

Falls are common in older adults and are a significant cause of mortality or fall-related injuries such as fractures, leading to reduced mobility and independence [1]. Given the increasing aging population, the occurrence of falls and healthcare-associated costs are projected to rise [1,2]. In fact, the aging process can lead to changes in some modifiable risk factors for falls. It is widely accepted that body composition changes, particularly a reduced muscle mass in the lower limbs and loss of bone mineral density (BMD), are major indicators of falls or fall-related fractures [3,4]. In addition, a decrease in physical function, such as a loss of muscle strength, and cognitive performance, particularly a slower processing speed, can enhance the risk of falling, especially in those with a history of previous falls [5,6]. In this way, it is essential to promote specific interventions to prevent the negative consequences of falls.

It is well established in the literature that single (e.g., exercise alone such as resistance training) or different combinations of interventions (e.g., exercise alongside vitamin D supplementation, or balance plus strength training) may prevent falls in community-dwelling older adults [2,7,8]. However, the intervention type, frequency, duration, participant’s mean adherence, or participant’s satisfaction level may influence the intervention’s effectiveness and should be investigated. Recent studies have shown an association between long-term exercise (at least 24 weeks, three times per week at a moderate intensity) and a reduction in the number of falls or fall-related fractures in community-dwelling older people [1,9].

Beyond physical function, exercise training leads to enhancements in cognitive function, such as processing speed [10,11]. The connectivity between physical activity/exercise and cognitive function is well established, and the potential mechanisms supporting the protective effects of exercise on cognitive abilities are described in the literature [12]. According to a previous study, this relationship can lead to hippocampal changes that promote neurogenesis and synaptogenesis processes through neuroplasticity. Concomitantly, positive effects of cognitive-based interventions (e.g., computerized cognitive training) on physical performance have been reported, leading to significant improvements in risk factors for falls, such as mobility, balance, and gait impairments [13,14]. However, an interactive cognitive–motor (ICM) intervention, promoting simultaneous cognitive and motor stimulation, may present better results in physical and cognitive functions, particularly in risk factors for falls, and should be preferred to a single intervention [15,16].

In this way, a psychomotor intervention program directed at older adults may present promising results for physical and cognitive functions [17,18,19] and can be considered an ICM intervention. Psychomotor therapy uses movement and corporality as the main resources to optimize physical, cognitive, affective, and perceptual skills through physical activity and functional body movements [17], in which reaching high-intensity training or performing high-impact exercises are not concerns. However, the potential effects of this therapy on body composition and physical and cognitive functions are still poorly known given the lack of studies, and its potential to reduce the risk of falls should be further explored.

On the other hand, whole-body vibration (WBV) training may improve bone mass and reduce the incidence of falls, thus minimizing fracture risk in case of a fall [3]. Moreover, WBV promotes muscle contractions by mechanical stimulation/oscillation and could improve physical function performance, particularly muscle strength, a critical risk factor for falls [3,20]. WBV can also improve some aspects of cognition [21]; nonetheless, little is known about the WBV effects on older adults’ processing speed.

Given the potential benefits of both interventions, we hypothesized that a combined intervention, including a psychomotor intervention and WBV training, could emerge as an effective and novel intervention to reduce the risk factors for falls or fall-related fractures. Additionally, few ICM programs have included a no-intervention follow-up [22,23], so the potential positive effects on body composition and physical and cognitive functions over time remain unclear. Thus, this randomized controlled trial (RCT) aimed to evaluate the effects of two ICM programs (psychomotor intervention versus psychomotor intervention + WBV) on body composition, lower-body strength, and processing speed in community dwellings at risk of falling.

## 2. Materials and Methods

### 2.1. Study Design and Participants

This 24-week RCT followed a single-blinded design and was performed between March 2018 and January 2019 (as described elsewhere [24]). Three groups were included: (1) experimental group 1 (EG1), which performed a psychomotor intervention program; (2) experimental group 2 (EG2), which underwent a combined program (psychomotor intervention program + WBV); and (3) the control group (CG), in which participants were asked to maintain their daily life routines. Participants were evaluated at baseline (m1), after 24 weeks of intervention (m2), and after a 12-week no-intervention follow-up (m3). After the follow-up evaluations, participants allocated in the CG were invited to participate in a fall prevention program. This RCT was reported according to the Consolidated Standard of Reporting Trials (CONSORT 2010) guidelines (http://www.consort-statement.org; accessed on 20 January 2021). In addition, a concise overview of the intervention programs was described according to the TIDieR checklist (https://www.equator-network.org/reporting-guidelines/tidier/, accessed on 20 January 2021). This study was registered at ClinicalTrials.gov (NCT03446352) on 26 February 2018.

The sample size was calculated using the online G*Power software, considering an effect size = 0.25 [25], alpha = 0.05, and statistical power of 95%. Hence, a minimum sample size of 45 participants was determined (15 participants for each group) to identify significant changes. The number of participants was increased to cover an expectable dropout rate. Thus, 61 community-dwelling Portuguese older adults were enrolled via verbal invitation and leaflets placed in community settings such as senior associations, recreation centers, and city halls.

Inclusion criteria required: (a) males or females aged 65 years or more; (b) score of ≥18 points (moderate or high physical functioning) on the Composite Physical Function scale [26]; and (c) a history of falls (≥1 fall) in the preceding six months or scoring 25 points or below (high risk of falling) on the Fullerton Advanced Balance scale [27]. Exclusion criteria were as follows: (a) scoring ≤22 points (cognitive decline) in the Mini-Mental State Examination (MMSE) [28]; (b) dependent mobility (walk without walking aids); (c) musculoskeletal (diagnosis of osteoporosis (T-score of −2.5 or below); recent lower-limb fracture; knee or hip prostheses), cardiovascular (pacemaker), and neurological (epilepsy) conditions that could compromise participants’ well-being [29]; and (d) participation in a regular exercise program over the last six months [30].

Among 61 candidates, 56 volunteers met the inclusion criteria (47 women and 9 men), and 5 volunteers were excluded, as described in Figure 1. After baseline evaluation, participants were randomly assigned according to simple randomization procedures with sequential numbers (1:1:1 ratio), performed by an investigator with no clinical involvement in the trial. The online “Random Team Generator” (https://www.randomlists.com/team-generator, accessed on 2 April 2018) was used, and participants were allocated into three groups: EG1 (*n* = 18), EG2 (*n* = 19), and CG (*n* = 19).

All the participants gave written informed consent. Ethical approval for the study was provided by the institutional research ethics committee on human health and well-being (reference number 16012), following the guidelines of the Declaration of Helsinki.

### 2.2. Procedures

The same trained rater, who graduated in rehabilitation sciences, conducted the participants’ assessments individually at the university laboratories. The evaluator was blinded to participants’ allocation. Cognitive tests and questionnaire completion were performed in a room with minimal noise and a comfortable temperature. Physical function and body composition variables’ assessments were undertaken in appropriate laboratories. Before each cognitive and physical assessment, participants were instructed with a verbal explanation, followed by a practice trial.

### 2.3. Outcome Measures

Body composition was assessed by dual-energy X-ray absorptiometry (DXA—Hologic QDR, Hologic, Inc., Bedford, MA, USA), which is considered a reliable, accurate, and safe imaging modality to measure changes in body composition and bone [4]. This assessment involved fat mass (%); lean body mass (kg); total bone mineral content (BMC) (g); total BMD (g/cm^2^); T-scores (*n*) as reference values for healthy young adults; and Z-scores as reference values for age and gender (*n*). Daily quality assurance was performed through a Hologic Spine Phantom.

Lower-body strength and muscle resistance were measured by the 30-s Chair Stand Test (30CST), in accordance with the methodology proposed by Jones, Rikli, and Beam [31]. The number of full and corrected stands in 30 s was recorded. Furthermore, the maximal strength of the knee extensors and flexors (60º/s; a range of motion of 90º) was assessed with an isokinetic dynamometer (Biodex System 3, Biodex Corp., Shirley, NY, USA), which was established as a reliable assessment device in community-dwelling older adults [32]. After a practice trial, one test trial was performed, including a set of three concentric repetitions. The highest peak torque value (N·m) reached in the test was recorded for further analysis.

Processing speed was assessed by the Trail Making Test (TMT) A and B, according to the instructions proposed by Cavaco et al. [33]. The time (s) to complete TMT-A and TMT-B was recorded as the number of errors.

Fall occurrence was assessed through an interview based on a script that comprises information about the date of each fall and the circumstances surrounding it (e.g., fall-related injuries, type, and location of fall). This oral interview was conducted to double-check for false-positive or false-negative responses. A fall was defined “*as an event which results in a person coming to rest inadvertently on the ground or floor or other lower level*” [34]. The self-reported number of falls was collected at baseline (retrospective falls over the previous six months) and post-intervention (prospective falls over the six intervention months).

#### Complementary Outcome Measures

To assess the exercise intensity, the Borg Rating of Perceived Exertion [35] scale was used, based on effort levels ranging from 6 points (very, very light) to 20 points (very, very hard) [36]. Participants’ satisfaction level was assessed by using the Caregiver Treatment Satisfaction questionnaire, which ranged between 1 point (extremely dissatisfied) and 5 points (extremely satisfied) [37]. Sociodemographic characteristics (age, sex, and educational level) were collected by means of an interview based on a script. The cognitive state was assessed by the Portuguese version of the MMSE [28]. Standing height (m) and body mass (kg) were measured through a stadiometer (Seca 206, Hamburg, Germany) and an electronic scale (Seca 760, Hamburg, Germany), respectively, and body mass index (kg/m^2^) was calculated. To assess the physical independence, the Composite Physical Function scale was used, which includes an ample range of functional abilities [26]; this 12-item self-report scale can range between 0 (worst) and 24 [10] points, and participants were categorized as “low functioning” (score: <18), “moderate functioning” (score: 18 to 23), or “high functioning” (score: 24). Each participant’s habitual physical activity was measured using the short version of the International Physical Activity Questionnaire (IPAQ) by means of the metabolic equivalent of task ([MET]-min/week), recording the time (min/day), the frequency (days/week), and MET intensity (i.e., walking: 3.3 MET; moderate: 4.0 MET; or vigorous: 8.0 MET). Physical activity was computed as the sum of metabolic expenditure spent on the three types of activity, each one calculated as time × frequency × MET intensity [38].

### 2.4. Interactive Cognitive–Motor Programs

Both programs were performed three times per week (75 min/session) on alternate days, with up to 10 participants in each class. All supervised sessions were delivered by the same specialist, who has a master’s degree in rehabilitation sciences, at the gerontopsychomotricity laboratory. Sessions were rescheduled for those who were absent for 3 consecutive sessions.

The ICM programs included cognitive and motor tasks [15]. Adaptative, specific, and progressive tasks were performed over the intervention period. These tasks followed the American College of Sports Medicine recommendations (e.g., gradual intensity/difficulty increase: initial stage comprising 2 sets of 8 repetitions and final stage comprising 3 sets of 15 repetitions) [39]. Physical exercises were executed using the participant’s body weight or affordable equipment such as fitballs, resistance bands, rubber mats, or unstable surfaces. A moderate exercise intensity (~13 points) on the Borg RPE scale was a target in both programs.

#### 2.4.1. Psychomotor Intervention Program

This program included the main principles of a psychomotor intervention directed at older people (e.g., body-mediated activities such as body scheme awareness) and was focused on ICM stimulation. Each class started with a 5 min beginning ritual and a 10 min warm-up. This phase involved join rotation (from neck to ankle) and a quick dual-task activity for neurophysiological activation (e.g., standing up and sitting down from the chair or pointing body parts according to arithmetic tasks). The main phase (50 min) consisted of different interactive activities (sensory/neuromotor exercises) that promote simultaneous cognitive and motor stimulation for alternate periods of approximately 15 min (i.e., the first 15 min comprised activities with greater cognitive demand, followed by 15 min with greater motor demand). The previous phase included neurocognitive activities (e.g., processing speed: select different animals/flowers based on relevant stimulus, as quickly as possible), motor activities (e.g., postural muscle and lower-limb exercises: dorsi-plantar flexion, such as standing on toes; knee extension/flexion, such as bodyweight squats), and dual-task paradigms (e.g., fitball wall squats simultaneously with a regressive countdown by 3 from 30 or while reciting their phone number backwards). During the 5 min cool-down phase, stretching exercises or relaxation methods using massage balls for body awareness development were performed. Lastly, at the 5 min finishing ritual, participants were asked to record their exercise intensity (RPE scale) and satisfaction levels (Caregiver Treatment Satisfaction questionnaire).

#### 2.4.2. Combined Exercise Program

As a complement to the psychomotor intervention program, participants in the combined exercise program were instructed to individually perform a WBV program (initial stage: 3 min; final stage: 6 min) on a side-alternating vibration device (Galileo^®^ Med35). Participants were asked to stand up on the platform without shoes while holding the handlebar with bent knees (~30° of knee flexion) and an erect trunk position to prevent musculoskeletal injuries. The exercise volume was also increased gradually during the 24-week intervention (exercise time: 45–60 s; the number of series: 4–6; and frequency: 12.6–15 Hz). An amplitude of 3 mm and a 1 min seated rest between series were always performed.

### 2.5. Statistical Analysis

All statistical analyses were conducted using the SPSS software package (version 24.0, IBM SPSS Inc.). According to the Shapiro–Wilk and the Levene test results, repeated measures ANOVA assumptions were not met. Thus, non-parametric statistics were performed. The Friedman test was used for within-group comparisons, and the Kruskal–Wallis test was used for between-group comparisons. Pairwise post hoc tests were also carried out when significant differences were found. Lastly, the Wilcoxon test was performed to compare paired fall data between the baseline and the post-intervention (i.e., number of falls).

Data are presented as means ± standard deviations or frequencies (%). The variation value was calculated between the baseline, post-intervention, and follow-up evaluations as ∆: moment_x_ − moment_x−1_. For significant differences between the evaluation moments, the respective delta percentage was also computed by the following formula: (∆%: [(moment_x_ − moment_x−1_)/moment_x−1_] × 100).

Effect size (ES) was determined for the within-group and between-group comparisons following the guidelines for non-parametric tests [40]. To quantify the practical meaningfulness of the treatment effect, the ES was computed as $r=(Z/\sqrt{N})$ and classified based on Cohen’s thresholds (small: 0.10; medium: 0.30; and large: 0.50) [41].

In all analyses, a *p*-value of <0.05 was considered statistically significant.

## 3. Results

Overall, 48 participants out of the 56 initially randomized completed the present study. Dropouts (dropout rate: 14.3%) were similarly distributed between groups, and participants who dropped out presented similar characteristics compared to participants who finished ICM programs (75 sessions each). Mean adherence was identical in both EGs (EG1: 82.3% vs. EG2: 84.3%), as were the exercise intensity (EG1: 12.9 ± 0.4 vs. EG2: 13.2 ± 0.3) and satisfaction level (EG1: 4.98 ± 0.3 vs. EG2: 4.99 ± 0.1). No adverse events from intervention programs were reported.

Table 1 summarizes participants’ general characteristics at baseline, and no significant between-group differences were observed.

Likewise, no significant differences between groups were found at baseline regarding body composition, physical function, or cognitive function variables.

Table 2 presents the findings of our study regarding the body composition variables. Within-group comparisons evidenced significant improvements from baseline to post-intervention evaluations only in the EGs, mainly in EG2. Specifically, the results showed that the programs induced improvements in the following variables: “Total BMC” (∆_m2−m1_% EG2: 11.4%, *p* < 0.001), “Total BMD” (∆_m2−m1_% EG1: 2.1%, *p* = 0.040; ∆_m2−m1_% EG2: 7.1%, *p* < 0.001), “T-score” (∆_m2−m1_% EG2: 46.0%, *p* < 0.001), and “Z-score” (∆_m2−m1_% EG2: 243%, *p* < 0.001). These results were not maintained at the follow-up evaluation, in which EG2 demonstrated a significant decreasing trend in the previous variables, namely, “Total BMC” (∆_m3__−m2_%: −6.9%, *p* = 0.002), “Total BMD” (∆_m3__−m2_%: −5.0%, *p* = 0.001), “T-score” (∆_m3__−m2_%: −72.2%, *p* = 0.001), and “Z-score” (∆_m3__−m2_%: −53.2%, *p* = 0.008). The respective effect sizes from baseline to post-intervention were medium (0.32) in EG1 and large (0.56 to 0.59) in EG2, whereas those between post-intervention and the follow-up were large (0.57 to 0.62).

Table 3 displays the analyses within and between groups for physical function concerning lower-body strength variables. Within-group comparisons between the baseline and post-intervention evaluations detected significant improvements in both EGs. In particular, the results showed that the programs induced improvements in the variable “30CST” (∆%_m2__−m1_EG1: 45.2%, *p* < 0.001; ∆_m2__−m1_% EG2: 42.9%, *p* < 0.001), representing an increase in the number repetitions. However, these improvements at the post-intervention were not maintained at the follow-up evaluation, with a considerable performance decrease in both EGs (∆_m3__−m2_% EG1: −21.4%, *p* = 0.001; ∆_m3__−m2_% EG2: −21.6%, *p* = 0.008). Additionally, significant differences among groups were also found at the post-intervention in this variable between EG1 and the CG, as the participants in EG1 achieved ~6 more repetitions than those in the CG (*p* < 0.001), as well as between EG2 and the CG, in which participants in EG2 executed ~5 more repetitions than those in the CG (*p* = 0.004). The within-group ES from baseline to post-intervention in EG1 (0.62) and EG2 (0.60) was large and remained large between the post-intervention and the follow-up (EG1: 0.63; EG2: 0.58). The ES between groups was also large between EG1 and the CG (0.69) and between EG2 and the CG (0.56). In regard to the maximal strength of the knee extensors and flexors variables, despite descriptive analysis suggesting an increase of 8.9% at post-intervention in the variable “Isokinetic peak torque (extension 60º)” in EG2, significant differences were only detected between the baseline and the follow-up evaluations in EG1 and the CG. A significant decrease between baseline and the follow-up was observed in the variable “Isokinetic peak torque (extension 60º)” in EG1 (∆_m3__−m1_%: −8.6%, *p* = 0.008, r = 0.31) and the CG (∆_m3__−m1_%: −9.2%, *p* = 0.008, r = 0.41) and in the variable “Isokinetic peak torque (flexion 60º)” in the CG (∆_m3__−m1_%: −12.9%, *p* = 0.040, r = 0.51).

Concerning cognitive function (Table 4), namely, the processing speed variables, significant within-group changes between the baseline and the post-intervention were observed in both EGs. The results revealed that the programs induced improvements in the variables “TMT-A time” (∆_m2__−m1_% EG1: −20.8%, *p* = 0.011; ∆_m2__−m1_ % EG2: −24.0%, *p* = 0.008) and “TMT-B time” (∆_m2__−m1_% EG1: −23.1%, *p* < 0.001; ∆_m2__−m1_% EG2: −22.9%, *p* < 0.001). The previously described values showed a better performance after the 24-week intervention by decreasing the time to complete the tasks. These improvements remained evident in both EGs between the baseline and the 12-week follow-up evaluations for the same variables “TMT-A time” (∆_m3__−m1_% EG2: −20.0%, *p* = 0.014) and “TMT-B time” (∆_m3__−m1_% EG1: −19.6%, *p* = 0.001; ∆_m3__−m1_% EG2: −17.0%, *p* = 0.040). The corresponding effect sizes (r) were large between the baseline and the post-intervention periods in both EGs (EG1: 0.55 to 0.62; EG2: 0.51 to 0.58), while those between baseline and the follow-up were large in EG1 (0.61) and medium in EG2 (0.43 to 0.45).

In terms of the fall occurrence, within-group comparisons from baseline to post-intervention periods showed a reduction in the number of falls of 44.2% in EG1 and 63% in EG2 (EG1: 1.13 ± 0.8 vs. 0.63 ± 0.7, *p* = 0.021; EG2: 1.19 ± 1.0 vs. 0.44 ± 0.7, *p* = 0.007), while the CG presented similar results and remained unchanged (1.13 ± 0.3 vs. 1.06 ± 1.0, *p* = 0.763).

## 4. Discussion

Overall, the present study results evidenced that both programs were accepted and well tolerated by participants. They effectively improved bone mass, which is essential to prevent fall-related injuries such as fractures. Despite an increase in BMD within EG1, EG2, which combined the psychomotor intervention and WBV training, led to additional benefits for more bone mass variables, namely, BMD, BMC, T-Score, and Z-score, with a large ES in all of these variables. Likewise, both programs effectively improved physical (lower-body strength) and cognitive (processing speed) risk factors for falls and injuries and decreased the fall rate. The improvements in these risk factors were clinically relevant, as they all had a large ES. After the no-intervention 12-week follow-up, the enhancements in bone mass induced by the programs were not maintained, particularly in EG2. Likewise, the physical benefits induced by both programs were reversed, unlike the cognitive function improvements, which were maintained, particularly within EG2. Our study is the second to evaluate the effects of a psychomotor intervention combined with WBV training and only the third study investigating the effects of a psychomotor intervention as a fall prevention program [18,24].

Regarding the adherence rate and tolerability, a few ICM studies have been carried out over 24 weeks, three times per week, in community dwellings. Along these lines, compared to our EGs, the 24-week study conducted by Boa Sorte Silva et al. [23] showed a lower mean adherence (83.3% vs. 70%). Predicting compensatory sessions in case of health problems may be an effective strategy for reducing absenteeism. Moreover, the exercise intensity of the RPE scale corresponded to the defined target (~ 13 points) and guaranteed that all participants performed all tasks during the intervention programs.

In regard to body composition, compared to the psychomotor intervention program, the combined intervention induced improvements in BMD and BMC, T-Score, and Z-score, with a larger ES in all variables. Thus, these improvements within EG2 were more visible at an osteogenic level than muscular strength and muscle mass levels, as described above, which could positively influence fracture risk. The vibration exposure could lead to a more effective stimulation of bone formation, increasing the BMD and BMC. Furthermore, these results suggest that adding only ~5 min per session of WBV training in a psychomotor intervention can lead to additional benefits. Given the lack of ICM studies focused on body composition changes, the comparison of our study with other studies is limited. Contrary to the present study, the 24-week study carried out by Marín-Cascales and colleagues [42] found a significant decrease in total fat mass, both in the WBV group and in the multicomponent program group (aerobic and drop jumps exercises), in postmenopausal women. These authors also found no changes in total lean mass or BMD in either group. The findings of the previous study regarding total lean mass are consistent with our study findings. The best method to improve muscle mass or lean body mass is still unclear, and future investigations are needed since muscle weakness increases the risk of falling [3,20]. Furthermore, it is interesting to observe that our psychomotor intervention with low material effort also achieved significant improvements in BMD. Thus, our psychomotor intervention can also be recommended as an effective therapy to minimize bone loss. Concerning the improvements in BMC, our study evidenced superior improvements to the multicomponent 24-month program conducted by Englund and colleagues [43]. In the previous study, their EG, which included strengthening, aerobic, balance, and coordination exercises, increased BMC by 3.5%, while our EG1 and EG2 increased it by 5.3% and 11.4%, respectively, despite only EG2 presenting significant improvements. Therefore, our EG2 could positively influence the prevention of bone demineralization. At the follow-up, these improvements were reversed, especially in EG2, suggesting the importance of non-cessation WBV training in body composition. These results were followed by normative data comparisons of T-score and Z-score variations, in which lower mean scores represent a lower bone density.

With respect to physical function, namely, lower-body strength, both programs induced similar improvements. This is an unexpected finding because WBV training has been referred to as an effective program for improving muscle strength, alone or combined with other programs [20]. Therefore, it would be expected that an intervention that combines WBV and a psychomotor intervention, including strength stimulation, would provide additional benefits in muscle strength compared to the psychomotor intervention alone. At the post-intervention, both EGs significantly increased the number of repetitions performed in the “30CST” (EG1: 45.2%; EG2: 42.9%), with similar effect sizes. These results support the findings in previous studies, such as Desjardins-Crépeau et al.’s [11] study, in which only mixed aerobic and resistance training combined with cognitive training led to an increase of more than 45% in the number of repetitions. Additionally, compared to the 12-week study conducted by Hsien-Te Peng and colleagues [44], our EGs achieved a more accentuated increase in the number of repetitions than their ICM EG, which improved by 10.1% (21.8 ± 6.9 vs. 24.0 ± 6.4). For the maximal strength of the knee extensors and flexors, despite an increase of 8.9% in the variable “Isokinetic peak torque (extension 60º)” within EG2, it was not significant. However, these results are in accordance with other ICM studies that presented an increase of 10.9% in the knee extension force after 12 months of intervention [45]. The fact that both programs included mostly resistance strength exercises could help to explain these results. Therefore, these results suggest that ICM programs designed for fall prevention should consist of resistance strength exercises. However, for enhancements in maximal strength, both programs should focus more on muscle strength and power exercises, possibly through plate-loaded machines. The sessions’ intensity level at the RPE scale should target values between 13 and 15 [45]. Nevertheless, the specificity of a psychomotor intervention, mainly oriented to corporeality and self-awareness, does not incorporate or reach these high intensities in a session. After the 12-week follow-up, improvements induced by both programs in lower-body strength, particularly in the “30CST” variable, were reversed. These findings are similar to those from Blasco-Lafarga et al.’s study [22], which developed an ICM program (strength + cardiovascular exercises under dual-task paradigms). These authors pointed out that the effects of detraining were more marked in muscle strength than in other physical function outcomes, with muscle strength being the physical function capability with more sensitivity to an intervention program and the respective detraining. Considering our intervention programs’ specificity, the results highlight the need for detraining periods to be less than 12 weeks, which is in line with recommendations in Blasco-Lafarga and colleagues’ study [22]. Another recommendation is to implement a home-based program including strength exercises, while the psychomotor intervention is not restarted.

Regarding the processing speed of our study participants, both EGs showed significant post-intervention improvements, with slightly larger effect sizes in EG1. Thus, the WBV training did not lead to additional benefits. Our results are consistent and superior to other ICM programs in community dwellings. After 24 weeks of an ICM intervention (resistance/balance training + computerized cognitive training), the participants (74.5 ± 3.8 years) of the study carried out by Sipila et al. [45] performed the TMT-A and TMT-B tests in less than 3.4% and 8.3% of the time, respectively; compared to the present study, our EGs executed the TMT-A and TMT-B in at least 19% less time. The specificity of the computerized cognitive training, which was initially supervised and, after some sessions, carried out individually and unsupervised, may explain these differences. An unsupervised ICM intervention (exergames under different postural conditions) was also carried out in the 16-week study conducted by Schoene et al. [16], and no significant improvements were observed in participants’ (82.0 ± 7.0 years) performance in the TMT-A (37.1 ± 19.2 vs. 32.8 ± 12.2 s) and TMT-B variables (110.9 ± 60.0 vs. 107.7 ± 47.7 s). Finally, the 12-week study carried out by Desjardins-Crépeau et al. [11] focused on an interactive program (stretching and toning exercises + dual-task training program) that significantly improved the processing speed by 15.3% in the TMT-A test, whereas no significant differences in the TMT-B variable were detected. Likewise, the previous study was supervised, and participants (73.2 + 6.3 years) also performed computerized cognitive training. Although prior studies have shown significant improvements in several domains of executive function, supervised ICM interventions, such as our programs, without resorting to computerized cognitive training can lead to additional improvements in information processing. Moreover, the diversity of group exercises proposed in our programs, as dual-task paradigms targeting the enhancement of specific cognitive domains and brain regions such as the prefrontal cortex, could help explain our study results. In this way, it is recommended that fall prevention programs have these characteristics. Thus, these findings must be interpreted with caution. Considering the effects of the programs’ cessation, the processing speed improvement induced by both programs was maintained at the follow-up evaluation, especially within EG2. These findings are in line with other studies. In the study of Blasco-Lafarga and colleagues [22], after 14 weeks of detraining, the executive function results showed a slight decrease. Therefore, cognitive function losses seem to be less sensitive to a detraining period. This is important because cognitive improvements, particularly in processing speed, directedly reduce the risk of falls and can attenuate the decline in physical function over ten years [6].

Lastly, a significant reduction in fall occurrence was observed in both EGs at the post-intervention, especially within EG2, which showed fewer falls. Despite the WBV training’s low frequency (15 Hz) within EG2 to ensure a safe intervention, the mechanical stimulation and higher muscle activation provided by WBV could lead to a larger protective effect of the combined program for falls. The psychomotor intervention for fall prevention conducted by Freiberger and colleagues [18] reported the fall occurrence over the previous six months at baseline and during the 12-month follow-up, and no significant differences were observed. Likewise, few ICM programs include the number of falls as the main outcome. The 16-week study carried out by Gschwind et al. [46], which included a virtual-reality intervention program, showed a decrease in the incidence of falls in EG (−68.0%). However, alongside the specificity of a virtual-reality intervention, the retrospective falls of the previous study were collected over the previous 12 months at baseline, so comparisons to our study should be interpreted with caution. One of the first studies to directly evaluate the effects of WBV training on falls also showed a significant decrease in the fall rate only in the combined 18-month program (multicomponent physical training + WBV). However, these results are difficult to compare to our study given the long-term intervention, exclusively postmenopausal women participants, and the higher frequency used (25–35 Hz) on the WBV [47].

Some considerations related to our study’s findings should be made, such as the recommendation that older people at risk of falling actively engage in ICM programs and the recommendation to improve the ICM program by combining the psychomotor intervention with WBV training to potentialize the benefits in physical and cognitive risk factors for fall and fall-related injuries. In the absence of the WBV platform, the single psychomotor intervention is widely recommended since this ICM program has also been shown to induce benefits in fall and fall-related injury risk factors, namely, the processing speed, lower-body strength, and BMD.

Future studies should include more psychomotor measures potentially linked with falls, such as the body scheme or knowledge of body part impairments. Furthermore, physiological assessments, such as collecting the brain-derived neurotrophic factor levels or an electroencephalogram to evaluate more precisely the effects of a psychomotor intervention on brain neuroplasticity, can also be incorporated. Regarding the strengths of the present study, we highlight the RCT design, which included a follow-up, and the intervention length. Our study also has some limitations. First, this study followed a single-blinded design. Second, the dropout rate (14.3%) was high; however, it was lower than in other interactive cognitive–motor fall prevention programs [16]. According to the G*Power software, the sample size remained sufficient to detect significant changes, which allows the generalization of the findings to the target population. Third, participants were not randomly assigned by gender (i.e., first females, second males). Fourth, nutritional supplementation such as vitamin D intake was not controlled, allowing more efficient calcium absorption to potentialize the impact of both programs on bone mass; however, the impact of vitamin D supplementation on BMD in older adults is still inconclusive [48]. Lastly, despite the predominance of female participants in our study, it was less than that presented in other studies [1].

## 5. Conclusions

Our results suggest that both interactive cognitive–motor programs were accepted and were well tolerated by participants. They effectively improved bone mass, particularly the combined program, which evidenced additional benefits in BMC, BMD, T-Score, and Z-score. Both programs positively impacted physical and cognitive risk factors for falls and injuries. Moreover, they decreased the fall rate, suggesting successful fall and injury prevention programs in community dwellings at risk of falling. Both the psychomotor intervention program and the combined program were shown to enhance the lower-body strength and the processing speed, with similar treatment effects. After the 12-week no-intervention follow-up, the bone mass and lower-body strength improvements were reversed in EG2 and in both EGs, respectively. However, the improvements induced by both programs in processing speed remained after the detraining period, particularly in EG2. These findings highlight the potential benefits of a psychomotor intervention program as a fall prevention program. In addition, the study findings evidenced that only ~5 min of WBV training enhanced these benefits, mainly due to its protective effect on bone and fall-related fractures.

## Figures and Tables

**Figure 1 ijerph-19-07117-f001:**
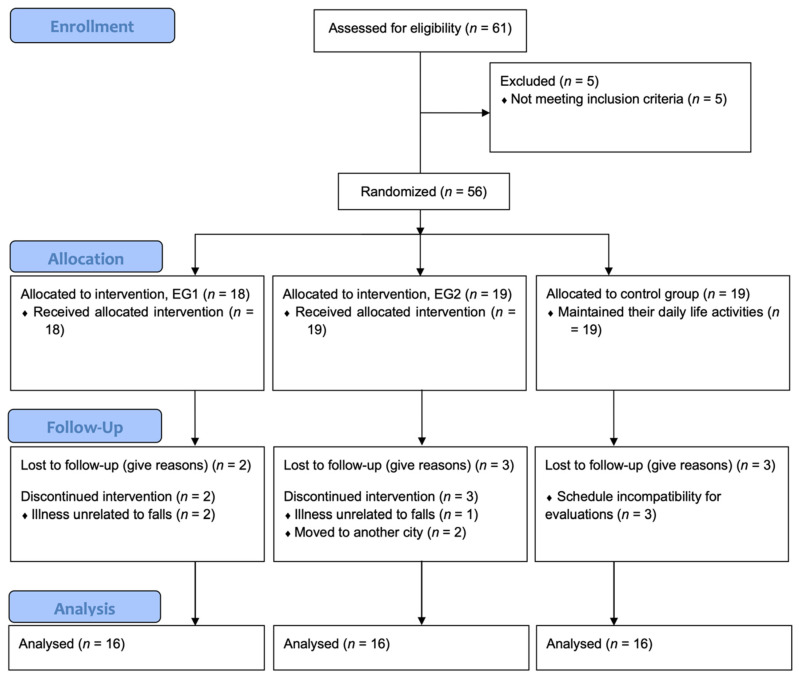
Flow diagram of the study participants.

**Table 1 ijerph-19-07117-t001:** General characteristics of the participants at baseline.

Characteristics	Prevalence or Mean ± SD	*p*-Value
Age (years)		
EG1	74.3 ± 5.4	0.750
EG2	74.7 ± 5.5	
CG	75.9 ± 5.7	
Sex, female (%)		
EG1	14 (87.5)	0.571
EG2	15 (93.8)	
CG	13 (81.3)	
Educational level (years)		
EG1	6.0 ± 2.6	0.992
EG2	6.1 ± 3.4	
CG	7.0 ± 5.1	
MMSE (points)		
EG1	27.7 ± 1.7	0.421
EG2	28.2 ± 1.7	
CG	28.4 ± 1.7	
BMI (kg/m^2^)		
EG1	29.1 ± 3.0	0.601
EG2	28.6 ± 4.3	
CG	28.0 ± 4.8	
CPF (points)		
EG1	21.5 ± 2.7	0.579
EG2	20.8 ± 2.2	
CG	21.4 ± 2.9	
IPAQ (MET-min/week)		
EG1	927.0 ± 557.9	0.803
EG2	953.4 ± 638.5	
CG	791.7 ± 482.2	
Number of falls within the last six months (*n*)		
EG1	1.13 ± 0.8	0.978
EG2	1.19 ± 1.0	
CG	1.13 ± 0.3	

Legend: SD, standard deviation; EG1, experimental group 1 (psychomotor intervention program) (*n* = 16); EG2, experimental group 2 (psychomotor intervention program + WBV) (*n* = 16); GC, control group (*n* = 16); MMSE, Mini-Mental State Examination; BMI, body mass index; CPF, Composite Physical Function; IPAQ, International Physical Activity Questionnaire; significant differences between groups, *p* < 0.05.

**Table 2 ijerph-19-07117-t002:** Impact of the interactive cognitive–motor programs on body composition variables.

		Baseline (A) (Mean ± SD)	Post-Intervention (B) (Mean ± SD)	Follow-Up (C) (Mean ± SD)	*p*-Value	Pairwise Comparison
Body composition						
Body weight (kg)						
	EG1	66.8 ± 9.7	67.5 ± 9.0	67.1 ± 9.1	0.494	--
	EG2	66.1 ± 10.4	65.7 ± 10.7	66.2 ± 11.2	0.223	--
	CG	67.9 ± 11.9	68.3 ± 12.0	67.2 ± 11.9	0.085	--
Fat mass (%)						
	EG1	39.3 ± 4.7	39.8 ± 5.1	39.0 ± 4.9	0.185	--
	EG2	41.1 ± 6.1	40.6 ± 6.2	41.0 ± 6.3	0.269	--
	CG	38.8 ± 6.9	38.7 ± 6.4	38.4 ± 6.7	0.570	--
Lean body mass (kg)						
	EG1	41.1 ± 7.1	40.9 ± 7.3	41.5 ± 7.3	0.368	--
	EG2	38.6 ± 5.6	38.6 ± 5.7	38.7 ± 5.9	0.829	--
	CG	40.2 ± 7.3	40.3 ± 7.7	40.3 ± 7.6	0.829	--
Total BMC (g)						
	EG1	1923.4 ± 313.0	2024.9 ± 402.0	1934.3 ± 271.6	0.047	--
	EG2	1705.9 ± 322.3	1901.0 ± 392.8	1770.3 ± 404.6	<0.001	B > A, C
	CG	1992.8 ± 443.0	1997.1 ± 485.0	2026.1 ± 461.7	0.939	--
Total BMD (g/cm^2^)						
	EG1	1.050 ± 0.098	1.072 ± 0.097	1.045 ± 0.091	0.022	B > A
	EG2	0.974 ± 0.112	1.043 ± 0.124	0.990 ± 0.133	<0.001	B > A, C
	CG	1.091 ± 0.141	1.084 ± 0.156	1.093 ± 0.146	0.570	--
T-score (*n*) *						
	EG1	−0.6 ± 1.2	−0.4 ± 1.1	−0.7 ± 1.1	0.062	--
	EG2	−1.6 ±1.2	−0.9 ± 1.2	−1.5 ± 1.3	<0.001	B > A, C
	CG	−0.6 ± 1.5	−0.7 ± 1.6	−0.5 ± 1.6	0.225	--
Z-score (*n*) *						
	EG1	1.3 ± 1.1	1.5 ± 1.0	1.3 ± 0.9	0.101	--
	EG2	0.3 ± 1.3	1.1 ± 1.3	0.5 ± 1.4	<0.001	B > A, C
	CG	1.4 ± 1.3	1.4 ± 1.4	1.5 ± 1.4	0.192	--

Legend: SD, standard deviation; EG1, experimental group 1 (psychomotor intervention program) (*n* = 16); EG2, experimental group 2 (psychomotor intervention program + WBV) (*n* = 16); CG, control group (*n* = 16); BMC, bone mineral content; BMD, bone mineral density; > significant differences within groups, *p* < 0.05; * these variables included a different number of participants per group due to limitations of reference population in DXA for gender and age in T-score (EG1: *n* = 14; EG2: *n* = 15; CG: *n* = 13) and Z-score (EG1: *n* = 13; EG2: *n* = 15; CG: *n* = 12).

**Table 3 ijerph-19-07117-t003:** Impact of the interactive cognitive–motor programs on physical function variables.

		Baseline (A) (Mean ± SD)	Post-Intervention (B) (Mean ± SD)	Follow-Up (C) (Mean ± SD)	*p*-Value	Pairwise Comparison
Lower-body strength						
30CST (*n*)						
	EG1	12.4 ± 3.2	18.1 ± 3.1 ^a^	14.2 ± 2.3	<0.001	B > A, C
	EG2	11.9 ± 3.5	17.1 ± 4.2 ^b^	13.4 ± 3.5	<0.001	B > A, C
	CG	13.2 ± 3.3	12.3 ± 3.2	12.0 ± 3.3	0.325	--
Isokinetic peak torque (extension 60°) (N·m)						
	EG1	82.3 ± 26.3	82.3 ± 25.6	75.3 ± 23.6	0.008	A > C
	EG2	71.2 ± 27.8	77.5 ± 21.0	75.6 ± 25.6	0.144	--
	CG	75.6 ± 24.9	71.7 ± 22.9	68.7 ± 19.7	0.010	A > C
Isokinetic peak torque (flexion 60°) (N·m)						
	EG1	42.5 ± 13.7	45.0 ± 14.2	43.3 ± 16.5	0.646	--
	EG2	40.3 ± 10.3	40.8 ± 9.5	39.9 ± 10.5	0.829	--
	CG	43.7 ± 14.7	38.7 ± 12.3	38.0 ± 11.3	0.022	A > C

Legend: SD, standard deviation; 30CST, 30 s Chair Stand Test; EG1, experimental group 1 (psychomotor intervention program) (*n* = 16); EG2, experimental group 2 (psychomotor intervention program + WBV) (*n* = 16); CG, control group (*n* = 16); > significant differences within groups, *p* < 0.05; ^a^ significant differences between EG1 and CG, *p* < 0.05; ^b^ significant differences between EG2 and CG, *p* < 0.05.

**Table 4 ijerph-19-07117-t004:** Impact of the interactive cognitive–motor programs on processing speed variables.

		Baseline (A) (Mean ± SD)	Post-Intervention (B) (Mean ± SD)	Follow-Up (C) (Mean ± SD)	*p*-Value	Pairwise Comparison
Processing speed						
TMT-A time (s)						
	EG1	91.3 ± 31.6	72.3 ± 27.8	85.1 ± 35.5	0.010	A > B
	EG2	85.2 ± 36.4	64.7 ± 29.3	68.2 ± 31.1	0.003	A > B, C
	CG	80.4 ± 39.8	73.3 ± 34.6	72.1 ± 30.8	0.305	--
TMT-A errors (*n*)						
	EG1	0.6 ± 1.1	0.3 ± 0.6	0.5 ± 1.0	0.438	--
	EG2	0.4 ± 0.5	0.3 ± 0.6	0.3 ± 0.6	0.368	--
	CG	0.4 ± 0.6	0.3 ± 0.6	0.4 ± 0.7	0.595	--
TMT-B time (s)						
	EG1	254.9 ± 70.9	196.0 ± 81.2	204.9 ± 81.6	<0.001	A > B, C
	EG2	224.0 ± 87.1	172.7 ± 76.9	186.0 ± 89.1	<0.001	A > B, C
	CG	202.5 ± 80.1	200.1 ± 83.1	187.8 ± 75.7	0.105	--
TMT-B errors (*n*)						
	EG1	2.1 ± 1.4	1.4 ± 1.2	2.0 ± 1.4	0.109	--
	EG2	1.6 ± 1.3	0.9 ± 1.1	1.3 ± 1.3	0.217	--
	CG	1.9 ± 1.3	1.4 ± 1.0	1.8 ± 1.2	0.234	--

Legend: SD, standard deviation; TMT, Trail Making Test; EG1, experimental group 1 (psychomotor intervention program) (*n* = 16); EG2, experimental group 2 (psychomotor intervention program + WBV) (*n* = 16); CG, control group (*n* = 16); > significant differences within groups, *p* < 0.05.

## Data Availability

The datasets used and/or analyzed during the current study are available from the corresponding author upon reasonable request.

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
