# Peer review of "Benefits of Two 24-Week Interactive Cognitive–Motor Programs on Body Composition, Lower-Body Strength, and Processing Speed in Community Dwellings at Risk of Falling: A Randomized Controlled Trial"

_ijerph, 2022, doi:10.3390/ijerph19127117_

Round 1

Reviewer 1 Report

Congratulations to the authors for conducting an interesting and relevant RCT. I have only minor comments:

- Please have English grammar carefully checked and corrected throughout the paper

- Methods: in addition to CONSORT, please consider reporting the methods using the CERT and/or TiDIER checklists available via the Equator Network website

Author Response

Response to Reviewer 1 Comments

Dear reviewer,

Thank you for your feedback and suggestions on our manuscript. We performed all changes (with track changes) in the manuscript.

Point 1: Please have English grammar carefully checked and corrected throughout the paper

Response 1: Thank you for the suggestion. The English language, grammar, punctuation, and spelling were carefully checked and corrected throughout the manuscript.

Point 2: Methods: in addition to CONSORT, please consider reporting the methods using the CERT and/or TiDIER checklists available via the Equator Network website.

Response 2: Thank you for the suggestion. In addition to CONSORT, we used the TIDieR checklist to report a concise overview of the intervention programs. We added information in the manuscript (lines 165-166; 280-283).

Reviewer 2 Report

The authors analyzed the impact of two ICM programs on some physical parameters and cognitive functions in community-dwellings at risk of falling. The obtained results suggest that the applied programs can successfully prevent falls, injuries and probably improve the quality of life in the examined population. Hence, minor revision as well as following corrections are recommended for this study:  

Line 29: Please explain the purpose of this statement! Why is it important for the abstract section?

Line 42: "Previous fallers" seems like an inappropriate term. A better manner of expression is needed here. 

Lines 43-44: A sentence cannot be started in this way. Additionally, this mistake is also repeated in the discussion section.

Lines 47-48: "Reducing fall risks" and "may prevent falls" have practically identical meanings.

Line 72: What do you mean by "performing an impact training"?

Lines 108-110: Please improve the readability and clarity of this statement. It is quite confusing now!

Line 116: How did you assess the mobility of the participants?

Lines 128-131: This statement should not be repeated twice.

Line 149: It would be useful for readership to explain the meaning of CST abbreviation. Additionally, what are T-score and Z-score? 

Lines 170-171: Did you mean "in the range of 6 points to 20 points"/or similar?

Lines 173-174: Which questionnaire was used to assess sociodemographic characteristics of the participants? Can you provide the reference, please.

Lines 192-196: Check this statement one more time, please! Some mistakes have been noticed here.

Lines 242-243: Check this sentence again, please!

Please insert the term "Legend" before explaining the abbreviation in all tables.

Line 270: Did you mean "in the following variables:"? Is this a better option in the context of this statement? 

Lines 273-274: "Were not seen" is an inappropriate expression.

Line 323: Are the values for EG1 missing in the first bracket? 

Lines 350-352: More that one reference should support mentioned statement!

Line 355: Did you mean "the 24-week study conducted/carried out by Boa Sorte Silva et al."? Similar mistakes have been observed several times throughout the discussion. 

Lines 500-503: This is an unnecessary statement. That has already been pointed out.

Lines 517-519: Check this statement one more time, please!

Author Response

Response to Reviewer 2 Comments

Dear reviewer,

Thank you for your feedback and suggestions on our manuscript. We performed all changes (with track changes) in the manuscript.

Point 1: Line 29: Please explain the purpose of this statement! Why is it important for the abstract section?

Response 1: Thank you for the observation. Some RCTs have this statement at the end of the abstract. However, since we already have this statement in the study design and participants section, we removed it from the abstract (line 28).

Point 2: Line 42: "Previous fallers" seems like an inappropriate term. A better manner of expression is needed here.

Response 2: Thank you for the observation. We changed for “… especially in those with a history of previous falls” (line 41). It is the term that is used in the referenced article.

Point 3: Lines 43-44: A sentence cannot be started in this way. Additionally, this mistake is also repeated in the discussion section.

Response 3: Thank you for the observation. We changed the start of the sentences for “In this way” and “Therefore”, respectively, in lines 42 and 662.

Point 4: Lines 47-48: "Reducing fall risks" and "may prevent falls" have practically identical meanings.

Response 4: Thank you for the observation. We removed the expression “…are effective in reducing fall risks” (line 59).

Point 5: Line 72: What do you mean by "performing an impact training"?

Response 5: Thank you for the observation. The statement was rewritten in the manuscript to clarify this point (lines 82-83).

Point 6: Lines 108-110: Please improve the readability and clarity of this statement. It is quite confusing now!

Response 6: Thank you for the observation. The statement was rewritten in the manuscript (lines 173-175).

 Point 7: Line 116: How did you assess the mobility of the participants?

Response 7: Thank you for the observation. We added information in the study design and participants’ section to clarify this point (line 181).

Point 8: Lines 128-131: This statement should not be repeated twice.

Response 8: Thank you for the observation. According to recent articles published at MDPI, this statement is placed in the methodology section and repeated in the declarations section. In this way, we will maintain both statements and wait for the editor’s instructions.

Point 9: Line 149: It would be useful for readership to explain the meaning of CST abbreviation. Additionally, what are T-score and Z-score?

Response 9: Thank you for the observations. We added the meaning of “30CTS” before the first abbreviation. T-score and Z-score are reference values with healthy young adults and with age and gender, respectively. We added this information in the manuscript (lines 226-227).

Point 10: Lines 170-171: Did you mean "in the range of 6 points to 20 points"/or similar?

Response 10: Yes. We changed the expression to “ranging from 6 points to…” (line 258).

Point 11: Lines 173-174: Which questionnaire was used to assess sociodemographic characteristics of the participants? Can you provide the reference, please.

Response 11: Thank you for the observation. The information described in the manuscript was not entirely accurate. The sociodemographic characteristics of the participants were collected by means of an interview based on a script that comprises information about age, sex, and educational level. We changed this information in the manuscript (line 262).

Point 12: Lines 192-196: Check this statement one more time, please! Some mistakes have been noticed here.

Response 12: Thank you for the observation. The statement was rewritten in the manuscript to clarify this point (lines 283-287).

Point 13: Lines 242-243: Check this sentence again, please!

Response 13: Thank you for the observation. The sentence was rewritten to clarify this point (line 365).

Point 14: Please insert the term "Legend" before explaining the abbreviation in all tables.

Response 14: Thank you for the suggestion. Done.

Point 15: Line 270: Did you mean "in the following variables:"? Is this a better option in the context of this statement?

Response 15: Thank you for the suggestion. Yes. We changed for "in the following variables:" (line 408).

Point 16: Lines 273-274: "Were not seen" is an inappropriate expression.

Response 16: Thank you for the observation. We changed to “were not maintained” (line 411).

Point 17: Line 323: Are the values for EG1 missing in the first bracket?

Response 17: Thank you for the observation. No. No significant differences were found between the baseline and the follow-up evaluations in the variable “TMT-A time”, in EG1 (as presented in Table 4). In this way, in the first bracket we only described the significant differences in the variable “TMT-A time”, within EG2.

Point 18: Lines 350-352: More that one reference should support mentioned statement!

Response 18: Thank you for the observation. We added another reference to the sentence (References 18 and 24; line 499).

Point 19: Line 355: Did you mean "the 24-week study conducted/carried out by Boa Sorte Silva et al."? Similar mistakes have been observed several times throughout the discussion.

Response 19: Thank you for the observation. Yes. We changed for “conducted/carried out” those references throughout the discussion.

Point 20: 500-503: This is an unnecessary statement. That has already been pointed out.

Response 20: Thank you for the observation. We removed this statement and moved some information (lines 718-720).

Point 21: 517-519: Check this statement one more time, please!

Response 21: Thank you for the observation. The statement was rewritten in the manuscript (lines 740-742).
